# The Impact of Angiotensin-Converting Enzyme-2/Angiotensin 1-7 Axis in Establishing Severe COVID-19 Consequences

**DOI:** 10.3390/pharmaceutics14091906

**Published:** 2022-09-08

**Authors:** Minela Aida Maranduca, Daniela Maria Tanase, Cristian Tudor Cozma, Nicoleta Dima, Andreea Clim, Alin Constantin Pinzariu, Dragomir Nicolae Serban, Ionela Lacramioara Serban

**Affiliations:** 1Internal Medicine Clinic, “St. Spiridon” County Clinical Emergency Hospital, 700115 Iasi, Romania; 2Department of Morpho-Functional Sciences II, Discipline of Physiology, “Grigore T. Popa” University of Medicine and Pharmacy, 700115 Iasi, Romania; 3Department of Internal Medicine, “Grigore T. Popa” University of Medicine and Pharmacy, 700115 Iasi, Romania

**Keywords:** renin-angiotensin-aldosterone system, RAAS, angiotensin-converting enzyme-2, ACE2, angiotensin 1-7, Ang (1-7), COVID-19, long-term effects

## Abstract

The COVID-19 pandemic has put a tremendous stress on the medical community over the last two years. Managing the infection proved a lot more difficult after several research communities started to recognize the long-term effects of this disease. The cellular receptor for the virus was identified as angiotensin-converting enzyme-2 (ACE2), a molecule responsible for a wide array of processes, broadly variable amongst different organs. Angiotensin (Ang) 1-7 is the product of Ang II, a decaying reaction catalysed by ACE2. The effects observed after altering the level of ACE2 are essentially related to the variation of Ang 1-7. The renin-angiotensin-aldosterone system (RAAS) is comprised of two main branches, with ACE2 representing a crucial component of the protective part of the complex. The ACE2/Ang (1-7) axis is well represented in the testis, heart, brain, kidney, and intestine. Infection with the novel SARS-CoV-2 virus determines downregulation of ACE2 and interrupts the equilibrium between ACE and ACE2 in these organs. In this review, we highlight the link between the local effects of RAAS and the consequences of COVID-19 infection as they arise from observational studies.

## 1. Introduction

Angiotensin-converting enzyme-2 (ACE2), a type I transmembrane protein expressed in many organs such as the heart, kidneys, lungs (type II alveolar cell), and intestine, plays a central role in the pathology of COVID-19 infection [1]. As it was in the case of SARS-CoV, the coupling of the virus molecule to the ACE2 receptor leads to fusion and downregulation [2,3]. Angiotensin-converting enzyme (ACE) and ACE2 are involved in the function regulation of several organs, metabolic condition, and, ultimately, the renin-angiotensin-aldosterone system (RAAS). Currently, this axis is described as two closely interlocking components: a deleterious arm—ACE—and a protective arm—ACE2 [4]. Despite the beneficial side of ACE2, a higher propensity for severe forms of infection has been previously assigned to higher levels of circulating ACE2 [5]. The vast majority of the effects precipitated by COVID-19 infection on key peripheral organs are linked to derangement of the ACE2/ACE axis. There is an ever-growing number of studies focused on observing the impact of the infection outside the lung. Additionally, the considerable amount of medical records allows for a thorough analysis of risk factors.

In this review, we aim to shed light on the molecular mechanisms underlying the tremendous consequences of COVID-19 infection, while keeping the RAAS at the heart of the discussion.

## 2. RAAS—The Classical View

The classical view of the renin-angiotensin-aldosterone system considers three peptides whose names comprise the acronym RAA axis. Angiotensinogen is converted to angiotensin I by renin, then further to Ang II by ACE.

Conventionally, the receptors angiotensin II binds to are only angiotensin receptor (AT) type-1 (AT1-R) and AT2-R. There have been other subtypes reported, such as AT3-R and AT4-R; however, the former has not been assigned a gene, so its existence is uncertain, while the latter is part of the AT4-R/Ang IV yet poses no affinity towards Ang II or its analogues (Figure 1) [6].

Currently, RAAS is viewed as a balance between, on one side, the detrimental actions of Ang II—AT1-R, and, on the other side, the protective effects of Ang II—AT2-R, Ang (1-7)—Mas receptor pathway, and Alamandine—Mas-related G protein receptor pathway. Stimulation of AT1-R subtype by the main peptide driver of RAAS—Ang II—leads to vasoconstriction, antinatriuresis, aldosterone level increase, sympathetic nervous system upregulation, and inflammation-mediated organ damage. Stimulation of AT2-R is generally considered to yield opposite effects, and is therefore protective [6]. The effects mediated by Ang (1-7)—MasR and Alamandine—MrgD are thoroughly discussed further in this text.

The degree of independence between the AT1 and AT2 receptors is highlighted by in vivo consequences after angiotensin-receptor blockers (ARB) administration [7]. Selective blockade of AT1-R would lead to a proportionally higher quantity of substrate binding to the remaining substrate, AT2-R. Furthermore, blockade of AT1-R at renal level results in increased levels of circulating renin and therefore higher levels of circulating Ang II, which would exhibit its protective effects on the only free receptors, AT2. Several studies have reported dramatically increased effects of AT2-R selective stimulation in the presence of small-dose ARB, which may be explained by a higher constitutive expression of AT1-R compared to AT2-R [8,9,10].

Metabolism end-products of Ang are Ang III and Ang IV. The former has a half-life five times lower in plasma than Ang II, and the latter exhibits its effects largely at the central level, with very low influence on peripheral AT1/2 subtypes [11]. For these reasons, Ang III and Ang IV contribute only slightly to the whole array of effects of Ang II.

## 3. RAAS—The Alternative Pathway

A new involvement in blood pressure control was unveiled in the early 2000s [12,13]. In this axis, Ang II is converted to Ang (1-7) by the catalytic activity of the then novel discovery, ACE2. The ACE–Ang II–AT1-R axis hyperactivation commonly leads to deleterious effects such as vasoconstriction, inflammation, endothelial dysfunction, thrombosis, and the well-established pro-hypertensive profile; the other, ACE2–Ang (1-7)–MasR typically reverts the forementioned effects. Ang (1-7) is found at a rather pseudo-stationary concentration state. There are at least three enzymes which aid in synthesis of Ang (1-7), and the reaction with the highest rate is different among the peripheral organs [14] (Figure 2).

### 3.1. ACE2 Enzyme

Following the efforts of discovering “ACE-like enzymes” in Drosophila melanogaster, mammalian homologue of ACE has been cloned and named “angiotensin-converting enzyme 2” (ACE2) [12,13,15]. ACE2 is an extracellular transmembrane protein expressed more restrictively to the surface of specific tissues including heart, kidney, endothelium, and testis [13]. Like ACE, it can be cleaved from the cell surface, resulting in the soluble form sACE2 [16]. Despite exhibiting considerable similarity and identity to human ACE, ACE2 functions exclusively as a carboxypeptidase, cleaving only the C-terminal residue from Ang I, yielding Ang (1-9) and from Ang II, likewise yielding Ang (1-7). Regardless, both the structures of ACE2 and ACE display a zinc-binding motif; typical angiotensin-converting enzyme inhibitors (ACEI) do not return any inhibition on ACE2 activity [12]. Arguably, inhibitors designed to act on C-terminal dipeptide-releasing enzymes—ACE—do not bind efficiently to C-terminal cleaving enzymes—ACE2.

### 3.2. SARS-CoV-2 and Its Receptor, ACE2

A full-fledged crisis emerged as the severe acute respiratory syndrome virus 2 (SARS-CoV-2) pandemics took over in late 2019. Most coronaviruses determine moderate enteric and pulmonary distress through the establishment of a proinflammatory profile [17]. SARS-Cov-2 belongs to the genus Betacoronavirus, but unlike the common coronaviruses, precipitates serious respiratory dysfunction [18,19]. ACE2 behaves as a receptor for SARS-CoV-2, and once the complex is formed, it is immediately internalized [20,21]. Surface expression of ACE2 is therefore diminished through consumption; circulating serum levels—sACE2—may also decrease, accounting for an absolute decrease in shedding. A decline in ACE2 levels will most probably be denoted by an alteration in synthesis of both Ang (1-9) and Ang (1-7). Furthermore, degradation of the hypertensive components of RAAS may be impeded by the low levels of enzyme ACE2. Therefore, a disequilibrium throughout the entire RAA axis is expected. While Ang (1-9) exhibits only few of the characteristics SARS-CoV-2 interferes with, Ang (1-7) is the most important substrate to account for the extensive effects of the infection.

### 3.3. Angiotensin (1-9)

Ang (1-9) is considered an intermediate product. The production step is catalysed by ACE-2, while consumption is catalysed by ACE. Ang (1-9) should exhibit in vivo a pseudo-stationary concentration; the absolute value of concentration at a specific time is directly determined by—amongst other factors—the ratio of the enzymatic activities of ACE-2 and ACE. Any alteration in ACE-2 activity would therefore yield significant effects on a systemic scale. The first direct receptor-mediated effect turned out to be the antihypertrophic effect mediated by AT2-R, independent of MasR pathway manipulation [22].

Infusion of Ang (1-9) in hypertensive elevated Ang II level rats resulted in significant blood pressure drop and circulating Ang II level reduction. A co-infusion experiment aimed to disclose the receptor for Ang (1-9) proved that MasR antagonist did not affect the beneficial properties of Ang (1-9), while AT2-R antagonist completely reversed the effects [22]. Controversial claims regarding either a possible interaction of Ang (1-9) with AT1-R, or in vivo transformation of this peptide to Ang II, were advanced in an attempt to explain the in vivo arterial pro-thrombotic activity exhibited by Ang (1-9) in rats [23,24]. To date, there are no reports of an enzyme to aid in conversion of Ang (1-9) to Ang II, nor a quantified affinity towards receptor subtypes other than AT2-R angiotensin. Extracellular signal-regulated kinase (ERK)1/2 pathway downregulation seems to be the key mechanism involved in the antifibrotic and antihypertrophic properties exhibited through AT2-R stimulation [25].

### 3.4. Angiotensin (1-7)

Angiotensin (1-7) is the most potent product of the protective arm of RAAS. Synthesis can take place three ways: (a) directly and most efficient, from Ang II by catalytic action of ACE2 [26]; (b) directly, from Ang I, catalysed by thimet oligopeptidase (THOT), neutral endopeptidase (NEP), and prolyl endopeptidase (PEP) [27,28]; and (c) through an intermediary product Ang (1-9), and therefore with a slow reaction speed, by means of ACE2 then ACE [29]. Degradation of Ang (1-7) takes place by parallel first-order kinetic reactions; decarboxylation yields a novel compound—Alamandine—by a not currently individualised enzyme. Interaction of alamandine with Mas-related G protein-coupled receptor (MrgD) explained some of the previously noted effects of Ang (1-7) which could not be assigned to Mas axis [30].

#### 3.4.1. Intracellular Signalling Pathways Employed by Angiotensin (1-7)

The majority of the effects generated by this heptapeptide are mediated by the MasR pathway. Downstream activation leads to Akt phosphorylation, upregulation of inducible nitric oxide synthases, and intracellular cyclic guanosine monophosphate (GMP) increase.

Besides the classical MasR pathway, Ang (1-7) mediates some of its cardioprotective effects by behaving as a beta-arrestin recruiter. Agonistic action at the active site of AT1-R leads to recruitment of beta1/2 arrestins, intracellular molecules which effectively block the active site in an irreversible manner. AT1-R engagement by Ang (1-7) leads ultimately to a transient activation of ERK1/2 axis, displaying a cardioprotective profile [31]. The affinity towards receptors other than MasR was also independently demonstrated by using human kidney (HK-2) cells [32]. Effects of low-concentration MasR agonist were not influenced by subsequent MasR blockade, but reversed by AT2 and AT1-R blockade, which owes a degree of affinity of Ang (1-7) to both receptors (Figure 3).

##### Influence on Metabolism

Activation of Glycogen Synthase Kinase-3 beta (GSK3β), a crucial insulin mediator, may be Akt-dependent or direct; nevertheless, metabolic effects should be expected after manipulation of Ang (1-7), and indeed, oral administration of coated peptide attenuates hyperglycaemia in type 2 diabetes mellitus rodents showing impaired insulin sensitivity [33,34]. While GSK3β further downstream controls glycogen synthesis, phosphorylation of Akt Substrate-160 (AS160) by Akt is required for Glucose Transporter Type-4 translocation to occur. GSK3β and AS160 synergically inhibit phosphorylation of the inhibitory site at Insulin Receptor Substrate 1.

ACE2 knockout mice showed reduced concentration of tryptophan and other large amino acids in blood (valine, threonine, and tyrosine) [35]. Histologically, the mice displayed diffuse mucosal inflammation and altered intestinal microbiota. Inflammatory susceptibility in ACE2-deficient subjects was remedied by supplying only tryptophan, which may indicate a benefit in a tryptophan rich diet in COVID patients. Along with the current knowledge regarding blood sugar levels and microbiota, insulin resistance and impaired glucose homeostasis can be partially explained by an imbalance of ACE forms [36].

##### Oxidative Stress

Nicotinamide adenine dinucleotide phosphate (NADPH) inhibition leads to reduced oxidative stress by inhibiting the detrimental activity of inducible nitric oxide (NO) synthase [37]. ACE2 knockout display hypertensive phenotypes aggravated by endothelial dysfunction, proved by the worse outcome induced by diabetes- and shock-induced kidney injury, viral lung injury, and chronic liver injury [38]. The unifying characteristic among these models is an increase in oxidative stress in ACE2 knockout male mice [39,40].

##### Arterial Blood Pressure

Vascular smooth muscle tone may be modulated either directly—as determined by upregulation of endothelial and neuronal NO synthase, subsequent cyclic GMP rise and vasodilation—or indirectly. Ang 1-7 treatment leads to normalisation of surface expression of the proteins Tumor Necrosis Factor (TNF)-Alpha Converting Enzyme (TACE) and Sodium Hydrogen Exchanger-3 (NHE3) in proximal renal tubule cells [41]. TACE is responsible for surface ACE-2 shedding; the high levels expressed in hypertension only compound the problem regarding low activity of Ang 1-7 [42]. NHE3 is a key transporter in proximal renal tubule cells; upregulation leads to increased reabsorption and drop in diuresis [43]. The reactive oxidative species (ROS) and NO generation in genetic deletion of ACE2 enabled researchers to observe the direct effect of Ang (1-7), and only slight but consistent elevation of blood pressure was noted [39]. Results are, however, discordant among different genetic mouse strains [44]. Vascular dysfunction—loss of the vessel ability to modulate local tone—only reinforces the previously noted consistently increased vascular tone [41].

##### Apoptosis

Mitogen-activated protein kinase pathway and nuclear factor kappa-light-chain-enhancer of activated B cells (NF-kB) transcription factor are both toned down in vascular smooth muscle cells, endothelium, and proximal tubule in kidney after infusion of the heptapeptide [45,46], resulting in a lesser inflammatory response, reduced cell death, reduced fibrosis, and satisfactory preservation of organ function. Finally, stimulation of MrgD by alamandine may account for the effects observed in triple blockade (MasR, AT1, AT2-R). Through phosphoinositide-dependent protein kinase 1 (PDK1), it mediates phosphorylation and activation of Akt, which downstream will activate inducible NO synthases. Figure 4 sums up the most important paths Ang (1-7) acts.

#### 3.4.2. Effects Mediated by Angiotensin (1-7) in Specific Organs

In order to assess the effects mediated by Ang (1-7) in an individual setting, several strategies have been employed to manipulate either the substrate or the receptor activities: pharmacological concentration alterations of circulating Ang (1-7), genetically modified expression of ACE2, and MasR. Significantly involved in the Ang (1-7)/MasR axis due to the high local Mas expression, the brain exhibits both local and systemic effects in response to manipulation of this axis.

##### Brain

Metabolism of the heptapeptide closely resembles the description in Figure 2. All components of RAS described above have been determined in brain tissue; however, not every compartment exhibits all of them, nor has any neuron type been identified to produce all those substances [47]. Despite that, Ang (1-7) presence has only been selectively demonstrated; ACE2 is present through every single brain compartment, mainly in neurons rather than glia cells [48]. MasR was identified in certain compartments mostly—but not exclusively—related to cardiovascular control: nucleus tract solitary (NTS), rostral and caudal ventrolateral medulla (RVLM, CVLM), paraventricular nucleus (PVN), and supraoptic nucleus (SON) [49].

Improved modulation of baroreflex by enhancing the bradycardic component, cardiac autonomic balance regaining, and renal sympathetic tone control restauration stand as explanation for the experiment in which long-term stimulation of Ang (1-7) axis leads to significant drop in hypertensive response to Aldosterone-NaCl artificial infusions or high-salt regime [50,51]. Aldosterone-NaCl treatment mimics the effect of Ang II systemically, while also downregulating any intrinsic renin or Ang II secretion. Deoxycorticosterone acetate (DOCA)-salt hypertension is predominantly characterized by upregulated sympathetic activity. Mas knockout mice (equivalent to ACE imbalanced profile) displayed—besides impaired baroreflex and aberrant renal sympathetic tone activity rise—blunted Jarisch-Bezold and chemoreceptor reflexes, with the entire neural network for blood pressure control being influenced [52].

The baroreflex response is the most intricate of all previously mentioned cardiovascular mechanisms. ACE inhibition led to improvement of bradycardic component in normotensive and spontaneous hypertensive rats (SHR). Inhibition of ACE2 axis by infusion of A-779 (an antagonist of Ang (1-7)) led to a significantly depressed sensitivity in normotensive, yet to a mild, almost insignificant attenuation of baroreflex in SHR, which comes to emphasize the imbalance between the two Angiotensin axis in the hypertensive subjects [53]. Infusion of A-779 in ACE inhibitor-treated SHR subjects promptly reverted the beneficial effects [54]. The favourable outcome met by ACEI treatment was most likely exhibited through Ang (1-7). Increased levels of Ang (1-7) at NTS—a key cardiovascular reflex command centre—and signalling through Phosphoinositide 3-kinase (PI3K) pathway may be responsible [55].

Fructose-fed rats treated with intracerebroventricular high-dose Ang (1-7) displayed protective (High-density lipoprotein) HDL-Cholesterol values, normal glucose tolerance, and normal levels of insulin. The peripheral cardioprotective and metabolic effects of central administration of Ang (1-7) were mediated via shifting Ang II/ACE2 equilibrium and further consequences on NO production rise (neuronal NOS activation via Mas axis) [56,57]. 

Emotional stress and anxiety-like syndromes were noted in Mas-deficient animals [58]. Chronic intracerebroventricular infusion of Ang (1-7) proved effective in reducing both anxiety- and depression-like syndromes [59]. Increase in ACE2 activity by use of ACEI showed a mood improvement in depression-suffering hypertensive patients [60]. The emotional response to stress can therefore be improved by administering ACEI class drugs, discontinuation of which would be deeply unwise from perspectives of both cardiovascular risk and cerebral involvement. Further benefit comes from the modulation of cardiovascular response evoked by acute emotional stress. Microinjections of Ang (1-7) into the basolateral amygdala—an area of the limbic system—lead to attenuation of pressor reflex in response to acute psychic stress [61].

##### Cardiac Direct Involvement

Coronary vessels seem to be favourably responsive at nanomolar order of concentration, with higher concentrations exhibiting no or constrictive effect [62].

Cardiomyocytes acutely exposed to Ang (1-7) only display a slight elevation in NO release by activation of endothelial and neuronal nitric oxide synthase (eNOS, nNOS) [63]; chronic exposure leads to significant effects on calcium-handling proteins, with increased expression of Sarcoendoplasmic Reticulum Calcium ATPase-2 (SERCA2), higher Calcium (Ca) transient amplitude and faster Ca uptake. This may explain the beneficial effects of ACE2 axis stimulation in chronic heart failure subjects [64].

Identification of Ang (1-7) and Mas axis in sinoatrial node provides the morphopathological fundamentals for the observed biphasic effect (an excessively high local peptide concentration worsened the prognosis) in administration of different concentration of Ang (1-7), only one order of magnitude apart [65,66]. Long-term overproduction of Ang (1-7) has proved to reduce cardiac fibrosis in many studies, reducing oxidative stress, autophagy prevention, and reducing mitogen-activated fibroblast proliferation [67,68]. The main axis involved are Mas-R, Insulin-like Growth Factor-1 Receptor (IGF1R)/PI3K/Akt, and alteration of mitogenic prostaglandin profile [69].

Contradicting its expected cardioprotective effects, overexpression of human ACE2 in mouse heart lead to ventricular tachycardia and sudden death [70]. An explanation may arise from the role of apelin, a peptide involved in a variety of cardiovascular pathological processes. Apelin serves as substrate for ACE2 and as a biomarker in cardiovascular diseases including coronary artery disease, stroke, ischemic heart disease, and infarction [71]. The lower the level of apelin, the worse the prognosis in those subjects. Manipulation of ACE axis towards a deep imbalance in favour of ACE2/Ang (1-7)/MasR may, therefore, prove detrimental in already fragile COVID patients.

##### Kidney

Inside the kidney water regulation network, Ang (1-7) and antidiuretic hormone (AVP) are involved in a very delicate balance. The effects of Ang 1-7 at central level may not parallel the ones on a peripheral level.

Ang (1-7) effects are distinct among different brain areas. Even in the same cerebral structure, there may be a whole range of distinctive effects owing to certain physiopathological conditions. As such, infusions in the PVN in rodents exhibit different effects according to the state of the subject involved. Following Ang (1-7) microinjection in a healthy subject, an increase in AVP was demonstrated [72]. However, an inverse relationship was noted between Ang (1-7) concentration and AVP release in haemorrhagic conditions of an ethanol-intoxicated model [57]. The underlying mechanism is the already consistent NO-induced release by Ang (1-7), which in the second study displayed an inhibitory effect on AVP secretion.

Ang (1-7) presents a wide array of effects on kidney function, which are way beyond our target. In almost all instances, perfusion/injection of Ang (1-7) in the microenvironment of the studied kidney (or glomerular portion) resulted either in no effect, or natriuresis/diuresis, antiproteinuric, or rarely Ang II antagonism [27]. Antidiuretic effects were noted in only two instances. Worth mentioning is the possible involvement of receptors other than Mas pathway; antidiuretic effects noticed in healthy normotensive rats could be blocked by pre-treatment with Angiotensin Receptor Blockers (ARB) Losartan—Ang (1-7) may display some affinity towards AT1R [73]. Additionally, antidiuretic effect in inner medullary-collecting tubule cells of water-loaded rats seems to involve V2-AVP receptor [74]. Administering A-779 prior to AVP blunted the water reabsorption; similarly, AVP antagonist forskolin prior to Ang (1-7) resulted in no increase in cyclic Adenosine Monophosphate (cAMP) and in no effect in the collecting tubule cell. Cross-antagonist administration following the inspected substance returned no inhibition; the mechanism underlying AVP/Ang (1-7) and their interconnecting substrate-receptor affinity may rely upon binding with subsequent cross-internalization of complex.

Renal vasculature in vitro exhibited afferent arteriole vasodilation in response to Ang (1-7) [75]. Recent in vivo studies prove the vasodilatory effect, yet also the inverse dependence on the degree of RAAS activation [76]; thus, low sodium intake and co-infusion of Ang II lead to hyperactivity of RAAS in the hypertensive human, which diminishes the beneficial effect of intrarenal infusion of Ang (1-7).

##### Vascular Actions

Ang (1-7) is endogenously produced by vascular endothelium, and therefore induces endothelium-dependent vasorelaxation. Furthermore, it enhances the vasodilator effect of bradykinin in several vascular beds [77].

Besides the well-known effect of vasodilation, Ang (1-7) exhibits antiproliferative and antithrombotic actions [78,79]. The antiproliferative side is mediated through an increase in prostacyclin (PGI2), which in turn reduced activity of mitogen-activated protein kinase (MAPK) Ang II-stimulated pathway. Neointimal thickness and stenosis reduction in rat stenting model, slowing of the osteogenic transition of vascular smooth muscle cells in calcified specimens, and cardiac fibroblast antiproliferative potential were noted [68,80,81]. The antithrombotic side is mediated by the increase in prostacyclin PGI2 and enhanced release of NO from platelets, both part of the Mas axis [82].

Most of the work around Ang (1-7) has been aimed at understanding its effect on vessels and blood pressure. While Ang II/AT1R is widely distributed in all vascular beds, Ang (1-7)/Mas is specific to certain vascular beds—kidney, lungs, adrenals, brain. In normotensive rats, either short or long-term infusion of Ang (1-7) resulted in increased conductance in the previously mentioned vascular beds. However, a proportional cardiac output increase allowed these subjects to not display any significant blood pressure changes [83].

Many of the vascular effects Ang (1-7) exhibits in vivo are mediated through Mas-R activation, which in turn activates eNOS and enhances NO release. Previously, we noted also the nNO inducting activity of the same Mas-R. Ang (1-7) is also involved in a fine-tuned network involving PI3K/Akt pathway, as demonstrated by the NO levels in Mas-transfected cells and in vivo transcription activation of Forkhead box protein O1 (FOXO1), a well-known negative regulator of Akt cascade [84,85]. The same PI3K/Akt pathway is—peripherally—involved in improvement of insulin sensitivity in fructose-fed rats [33].

In human endothelium, Ang (1-7) counteracts effects mediated by Ang II through a common enzyme, Src homology-2 domain-containing protein tyrosine phosphatase-2 (SHP-2). Antioxidant action occurs by opposing the activation of NADPH oxidase by AT1-R, while anti-inflammatory activity occurs through inhibition of NFkB nuclear factor translocation in nuclear cells and attenuation of Vascular Cell Adhesion Molecule-1 (VCAM-1) expression, otherwise an early marker of endothelial dysfunction [86,87].

Restoration of ACE2 function in stroke-prone spontaneously hypertensive rats (which constitutively present a low ACE2 level) leads to significant stroke risk decrease, lower blood pressure profiles, and improved endothelial function [88,89,90]. Function restoration and perhaps overshooting the imbalance of ACE axis towards the protective side may yield strongly beneficial therapeutic results.

## 4. Medium and Long-Term Complications Determined by Infection with SARS-CoV-2

### 4.1. Brain

Neurological involvement has been recognized in more than 80% of severe cases of SARS-CoV-2 infection [91]. The choroid plexus cells proved to be the sense and relay station for neuroinflammation in the context of SARS-CoV-2 infection [92]. Additionally, persistence of the virus for several months in various anatomic sites other than the lungs has brought the brain into focus [93]. There is a wide range of central nervous system (CNS) severe complications in infected patients, generating five pathological profiles: encephalopathies, inflammatory conditions, ischemic strokes, peripheral disorders, and other miscellaneous neurological impairments [94].

A constant clinical finding which emerges even before the onset of respiratory symptoms is the distress in smell and taste sensations in patients infected with SARS-CoV-2 [95]. A decrease of the sensory input leads to a corresponding decrease in the grey matter of the relay and final cerebral stations [96]. The olfactory system encompasses connections to and from the piriform cortex, hippocampus, parahippocampal gyrus, entorhinal cortex, and orbitofrontal cortex [97]. Therefore, it is sensible to put more effort into analysing the previously mentioned regions from whole brain slices. Significant alteration of any of those components (piriform cortex, hippocampus, parahippocampal gyrus, and orbitofrontal cortex) may indicate a direct effect of the infection while simultaneously reducing the error arising from cerebral degeneration owing to pre-existing conditions.

Computer Tomography (CT) and Magnetic Resonance Imaging (MRI) are the most useful imaging techniques for cerebral tissue. A longitudinal analysis employing images both prior and after infection would reinforce the direct correlation between brain slice aspect and long-term disease consequences [98].

Multimodal brain imaging of 401 patients both before and after infection with SARS-CoV-2 allowed researchers to gain more insight into the long-term effects of mild-to-moderate infection [97]. Case-versus-control analysis of brain slices returned significant longitudinal changes in several olfactory-related areas; among the 10 most significantly altered areas were the amygdala, the hippocampus, and the piriform cortex. An imbalance in the ACE2/ACE axis in the context of SARS-CoV-2 infection, along with important staining in the same mentioned areas for the Mas receptor, may explain the observed longitudinal alteration (persistent relative decrease of function) of the hippocampus, piriform region of the frontal lobe, and the amygdala [49,99].

A critical role of the RAAS is represented by the involvement in learning and memory control; long-term potentiation of limbic structures, particularly the hippocampus and amygdala, is mediated through the Mas axis [100]; the imbalance in the ACE axis will therefore disturb the stress response, learning, and memory through a dual coupled mechanism—organic volume density and function. Indeed, studies have already been advanced regarding COVID-19 infection and mental health [101,102]. Neurocognitive symptoms have been consistently mentioned, such as delirium, chronic attention span worsening, and decreased memory capacity. A key aspect regarding mental health is represented by the social burden the pandemic has brought along [101]; the social aspect in a patient with already altered mental health may be more significant than previously expected.

A worsened stress management, owing to the transformations occurred in the amygdala and hippocampus, will also have peripheral consequences, which will most probably parallel the direct effects of the Ang (1-7) level decrease on specific organs. Cardiovascular response to acute stress has been linked to local concentration of Ang (1-7) in amygdala [61].

Cognitive impairment is a characteristic for post-COVID patients, the older ages exhibiting the most significant difference when compared to uninfected subjects [98]. Recently, a correlation has been established between an increase in the protective arm of the RAA axis and cognitive function and depression attenuation [103,104]. An imbalance of the ACE axis, with the prevalence of the deleterious arm, may account for the overwhelming reports of depression in post-COVID patients [102,105,106,107].

### 4.2. Heart

Acute cardiac injury was commonly reported as a complication, and was 13 times more often in Intensive Care Unit (ICU) patients than in non-critical cases [108]. Several studies reported a direct correlation between cardiac Troponin T (TnT)—mirroring acute cardiac injury—and inflammatory profile markers (C-reactive protein CRP, leucocytosis, dimer, and procalcitonin) [109,110]. Anatomopathological analysis of tissue in non-survivors returned macrophage as the key cell owing to the local inflammation [111]. The category of patients with underlying cardiovascular disease (CVD) displayed a higher frequency of elevated TnT levels compared to patients with no history of CVD [109]. The same group of patients recorded a higher risk ratio of major complications—including arrythmias—and a higher infection severity. As such, TnT level could stand as the link between acute cardiac injury and life-threatening arrythmias, two major cardiac complications of COVID-19 infection.

The propensity for cardiac injury in severe forms of SARS-CoV-2 infection has been positively correlated to elevated Troponin I (TnI) levels immediately after hospital admission [112]. In addition, normal TnI levels in the first 24 h after admission are significantly linked to lower mortality [113].

Heart failure—quantified by N-Terminal pro B-type Natriuretic Peptide BNP (NT-proBNP)—was also in a direct relationship with inflammation and injury markers [109,110]. It is not yet clear whether this manifestation is due to pre-existing left ventricular dysfunction or is an acute phenomenon in the context of myocarditis. Apart from ACE2 involvement, lung injury may explain heart failure; drop in functional residual volume results in pulmonary hypertension and acute right heart failure.

Arrhythmias were observed significantly more frequently in patients presenting acute cardiac injury, and the incidence of the two complications were statistically correlated. Up to 44% of patients transferred to ICU displayed cardiac arrhythmia at some point during admission [114]. Although both ventricular and atrial arrhythmias were noted, the former may be the first apparent [115].

### 4.3. Renal

The pooled incidence of Acute Kidney Injury (AKI) in USA and Europe was 28.6%, and only 5.5% in China, the difference being represented by the definitions used [116]. With increasing severity of the disease, the proportion of AKI cases increases too. Physiopathology of AKI is an interlocking blend of local and systemic inflammation, endothelial injury, activation of coagulation pathways, and RAAS imbalance.

Anatomopathological findings reveal a more severe decline in renal function compared to actual necrotic process in the nephron tubule—characteristic decoupling for septic shock related AKI [117]. Other analysis returned thrombotic microangiopathy besides acute tubular injury and collapsing glomerulopathy [118]. The latter is the first clue in favour of endothelial activation and pro-thrombotic local status. Collapsing glomerulopathy, named, COVID-19 associated nephropathy (COVAN), is most likely explained by the same mechanism underlying HIV-associated nephropathy. The mechanism involved is represented by podocyte injury—reinforced by presence of viral particles inside cells—and filtration membrane disruption with subsequent proteinuria [119].

Endothelial dysfunction plays, again, a key role in pathology. Worth remembering is the association of elevated inflammatory markers with the poor prognosis of COVID-19-infected patients. Micro- and macro-vascular thrombosis in the context of endotheliitis have been thoroughly reported, including in the kidney vascular system [120,121]. Platelet activation, microvascular vasodilation, enhancement of immune cell adhesion (assembly of Neutrophil Extracellular Traps (NETs)), and increased vascular permeability all come to reinforce the pro-thrombotic status of a COVID-19 patient [122]. Patients at high risk of severe COVID-19 present with hypertension or diabetes, which themselves depict chronic endothelial dysfunction; reduction in bioavailability of NO may be a central element in explaining the severe course in these patients.

Sepsis-like injury may be explained by the direct action of inflammatory mediators such as TNF at renal endothelial and tubule cell level [123]. Interferon upregulation by the viral infection may represent the (or one of the) cytokines involved in podocyte injury-related proteinuria [124]. Complement activation also plays a pivotal role in endothelial dysfunction and AKI persistence and progression to fibrosis/chronic kidney disease. Higher levels of anaphylatoxins have been constantly recognized in COVID-19 patients, which led to a greater binding of C5a to its receptor on both endothelial (EC) and tubule epithelial cell [125].

### 4.4. Vascular

Reports from hospitalized patients revealed an imbalanced and exhausted immune profile, an apparent hyperactivation consistent with “cytokine storm”, macrophage activating syndrome, most severe in sickest patients. In these cases, immunoparalysis is characterized by significant drop in Human Leukocyte Antigen (HLA)-DR molecule expression on CD14 monocytes, and CD4 and Natural Killer (NK) cell cytopenia inversely correlated with IL-6 and CRP levels. IL-1β levels were lower than expected for such a tremendous inflammatory response [126].

EC activation occurs as response to IL-6, IL-1 (secondary to IL-6), damage- and pathogen- associated molecular pattern (result of viral-mediated cell death). Activated ECs exhibit pro-inflammatory gene expression, immune cell adhesion and chemotaxis, increased vascular permeability, and alteration of local thrombotic potential.

Kawasaki-like diseases (KD) coexisting with multisystem inflammatory syndrome in children (MIS-C) were reported in several instances of COVID-19 infection [127,128]. It seems that acute vasculitis in children may progress towards giant coronary artery aneurysms significantly more frequent in COVID-19 affected patients than in the KD patient, pre-COVID-19 era [129].

Pulmonary ECs dysfunction surely represents part of the picture of severe hypoxia and acute distress syndrome by altering the air–blood barrier thickness and permeability. Thrombosis in ECs activation is mediated both by EC secretion of Plasminogen Activator Inhibitor-1, Tissue Factor, von Willebrand factor, and NETs formation. Several reports returned up to 25% incidence of venous thromboembolism, with correlation between thrombotic events and lack of prophylactic anticoagulation [130,131]. Macrovascular thrombosis evidence is brought by a case series of ST-segment elevation myocardial infarction patients, but with no sign of plaque rupture at angiography [132]. EC dysfunction may contribute along the EC activation to the potential for thrombosis; knockout ACE2 rodents demonstrated EC dysfunction, and this has been recently linked to the thrombotic events susceptibility [39,133] (Figure 5).

## 5. COVID-19 Infection Severity: Proven Risk Factors and Their Link to ACE2 Expression

### 5.1. Age

COVID-19 infection plot for percentage of critical care cases (out of seropositive individuals) vs age displayed intriguing and oddly inconsistent behaviour. While infection was most mild in children, its severity escalated with age, particularly in the above-80-year-olds group.

It is currently known that children infected with COVID-19 have either no or mild symptoms. Depending on the study design and therapy reporting, asymptomatic patients represented from 16% to 36% [134,135,136]. Again, these rates are fairly underestimated due to the low detection in the asymptomatic category. No significant sex difference was noted, and no statistically relevant age distribution profile was noted after examining several studies [134,135,136,137,138,139]. There is conflicting evidence regarding risk factors for contracting the infection: obesity, chronic respiratory conditions, viral co-infection, and immunodepression [140]. Interestingly, though, the highest rate of patients to require critical care was consistently found in the lowest age groups [134,141,142,143]. Evidence was, however, difficult to find, due to most authors not linking cases which required invasive treatment to age groups. However, rates of both contracting the infection and requiring ICU were systematically reported as lower than in adults [138].

Several articles released in 2020 reporting COVID-19 cases in <19-year-olds have highlighted that young frail patients are at risk for developing severe form of infection [134,142,143]. In all cases, most patients demanding ICU or assisted ventilation were under 1 year-old. Furthermore, the category of patients 1 month-old or less were reported to be most vulnerable [134]. Independent risks under multivariable analysis were age, male sex, respiratory symptoms at presentation, and pre-existing medical conditions. Median interquartile range (IQR) age in patients requiring intensive care versus all admitted cases showed a shift towards lower values, presumably due to the high proportion of under 1 year-old subjects considered in this study.

A sensible theory to explain the peak in severity profile in infants advances the higher serum expression of ACE2 receptors in children owing to the higher predisposition of this age category to infection. Serum concentration of ACE2 in infants <1 year-old were reported to be significantly higher than any other children and adult age group [5]. This was, however, the only statistically significant finding of the authors, other age groups reporting similar quantities of sACE2. Availability of the pathogen’s receptor in the bloodstream may explain the higher symptomatic rate and mortality in the <1 year-old age group. It is worth noting the inverse relationship between ACE2 expression and infection intensity, which is particular only to infants. The biphasic role of ACE2 expression in tissues vs plasma has been noted in previous outbursts of SARS infections [144]. Nevertheless, recent work has produced conflicting evidence regarding sACE2 levels and patient prognostic [145,146].

Age-stratified mortality and ICU necessity rates reveal a steep profile with increasing age; considering that most of the infections in children and young adults go unnoticed, the steepness increases even more dramatically towards the elderly group. Many countries unfortunately became understaffed during the first peak of the pandemic in May 2020, so an in-hospital death rate drop in the highest age categories may observed.

Several studies have separately discussed risk factors for contracting the infection, while others have focused on disease progression. However, it was essential to establish a strong link between risk factors for development of disease, and severity or progression of infection once it was attained [140].

### 5.2. Demographic Factors

Age and male sex proved to be independent risk factors both for contracting the disease and admission to ICU [147]. Intriguingly, though, a study from the same year (2020) highlights a more in-depth method of assessing a patient and predicting individual risk of developing a more severe form of infection [148]. A correlation was advanced between biological aging—which in turn takes into consideration up to 9 biological markers—and severity of disease. Ethnicity as an independent risk factor returned inconclusive results; the lower socioeconomic status may be responsible in this analysis.

### 5.3. Arterial Hypertension

Hypertension was a constant risk factor for severity and mortality for in-hospital patients. It was determined that hypertension along with high average systolic blood pressure and high systole-diastolic variability (not individually) in COVID-19 patients lead to a darker prognosis compared to patients with low and stable blood pressure [149]. Hypertension alone may not be at fault. Advancing in age results in an imbalance between ACE/Angiotensin II and ACE2/Angiotensin (1-7) enzymes. Indeed, curvilinear association between ACE2 levels and age was noted until 55 years old, beyond which the circulating level of enzyme moderately dropped with advancing age [150]. The imbalance between the Ang II and Ang (1-7) axis is easily reinforced by the baseline pro-inflammatory status consistently present in older ages [151].

Shortly after the beginning of the pandemic, ACE2 was proven to be the principal receptor through which the virus gets inside the cell and then the bloodstream. Anti-hypertensive drugs belonging to the class of RAA axis inhibitors are commonly known to upregulate levels of ACE2, although evidence is scarce and disparate [152,153]. It may seem counterintuitive to continue medication and enhance the virus’s natural ability to penetrate in the body by over-expressing the virus receptor.

Still, patient management revealed that poor blood pressure control, both previously and during admission, lead to high mortality. Adequate hypertension management was still necessary. To support this hypothesis, evidence shows that previous hypertension treatment in COVID-19 patients may indeed lower the in-hospital mortality [154]; this surprising result was recognized after observing the outcome of hypertensive patients with a history of at least 6 months of treatment with ACEI/ARB inhibitors. Their conclusion was not, however, supported by similar subsequent studies, yet practitioners were reassured there was no reason to switch to other medication class or discontinue their patients’ current hypertension treatment [155].

### 5.4. Diabetes

Poor blood sugar control (HbA1C) comprised a significant risk factor for worse infection progression, but not for contracting it. It is demonstrated that diabetes patients present an inadequate immune response to infections: impaired cytokine signalling (impaired IFN-alpha synthesis) and high cellular response to molecular signals (highly sensitised monocytes) [156,157]. Concisely, the diabetes patient is vulnerable to fungi and bacterial infections, and shows an unbalanced response to viral agents. Moreover, as a response to ACE upregulation in different tissues, ACE2 enzyme is overexpressed in serum, liver, and pancreas of the diabetic patient, and may partially explain the higher vulnerability to COVID-19 infection [158].

### 5.5. Obesity

Poor weight management was the third most incriminated risk factor for mortality. Increased plasma ACE2 were positively correlated with body mass index (BMI) and glycosylated haemoglobin (HbA1C), thus predicting the high risk. A typical obese patient presents with the following cohort of diseases: hypertension, insulin-resistant diabetes, and dyslipidaemia [150]. All these factors considered, weight-related ACE2 level may represent too small a fraction of the whole risk to an obese patient.

### 5.6. Pregnancy

Out of the infected female patients, a significantly higher percentage of pregnant women required hospitalization compared to non-pregnant subjects [159]. The most likely mechanism for the higher susceptibility is represented by the relative immune suppression characteristic to pregnancy state. Increased infection severity in pregnant subjects positively correlated with higher age, greater weight, diabetes, and hypertension [160,161]. Mortality cases were ascribed to respiratory failure and multiple organ failure [162]. The increased susceptibility for acute respiratory distress syndrome may be linked to the alteration of forced vital capacity during pregnancy [160]. Moreover, a synergistic yet deleterious effect may be caused by the pro-thrombotic state common in COVID-19 patients and the hypercoagulable state in pregnancy; the resulting coagulopathy could represent the foundation of multiple organ failure [163].

## 6. Long-COVID Organ-Specific Management

In spite of the highly assertive vaccination campaigns, there were tremendous numbers of cases, increasingly higher with each wave of the pandemic. The last wave of the pandemic proved to be the most challenging, owing to the new category of "long-haul” COVID-19 subjects [164]. In some individuals, following the acute phase, the virus resides in specific tissues. Reactivation of the virus, accelerated by a prolonged immune dysfunction and a whole range of other systems degradation, may explain the diversity of long-lasting symptoms [165]. Specific follow-up for COVID-19 patients who developed at least one acute complication should be mandatory. This strategy may allow healthcare providers to identify certain long-term complications—found under the term “long-COVID”—in the early phases, and therefore the patients to benefit sooner from adequate treatment.

### 6.1. CNS Complications Evaluation

Depression-like syndrome, anxiety, and post-traumatic stress disorder (PTSD) are some of the most encountered psychiatric findings in COVID-19 survivors [107,166]. Severity of long-COVID syndrome is unlikely to be related to severity of the infection [167]. Precise identification of COVID-19 survivors who are prone to psychiatric sequels becomes extremely challenging, owing to the wide array of mechanisms through which COVID-19 affects the CNS and the neurological burden a pre-existing illness may exhibit [168]. Independent risk factors for long-COVID “brain fog” were female sex, severity of respiratory symptoms, and admission to ICU during hospitalization [169]. Indeed, acute lung injury has been demonstrated to yield significant long-term neurological impairment, hypoxemia representing a major mechanism [170]. There is important inconsistency regarding the time interval between when the screening for newly onset psychiatric disorders should take place and the infection symptom onset [166,169,171].

Magnetic resonance imaging of the brain proved a very powerful tool in assessing brain damage associated with COVID-19 infection [98,172]. The longitudinal effects were thoroughly documented, allowing for a deeper understanding of the link between olfactory sense loss and brain impact. It is worth remembering that psychiatric disorders may or may not be mirrored by an organic dysfunction. PTSD, as one of the most severe forms of psychiatric sequels in COVID-19 survivors, proves a model for difficult neurological assessment [168]. Differentiating cognitive complaint as belonging to psychological trauma or neurological insult is key to diagnosis of PTSD; however, the healthcare provider must always be aware that the patient they are examining may be displaying a combination of the forementioned. In addition, patients found in high-risk categories for infection may already display a varying degree of neurological impairment secondary to underlying disease. Even COVID-19 survivors who do not exhibit the whole myriad of symptoms characteristic to PTSD—the condition known as post-traumatic stress-disorder—may display subjective and objective cognitive deficits [173].

Therefore, assessment of psychiatric damage in the context of COVID-19 should occur earlier during treatment, target vulnerable categories of patients, and comprise a wider variety of symptoms, and must lower the threshold for diagnosis [168]. Further management of patients is limited only to pre-existing guidelines, but further COVID-19-related pathology should be suspected and the treatment tailored accordingly.

### 6.2. Cardiovascular Complications Evaluation and Guidance

Onset or persistence of cardiovascular symptoms weeks after the acute infection phase requires the patient to undergo an extensive battery of tests tailored to the main presumed diagnosis. Physical examination should be focused on the patient’s symptoms and signs, and the common cardiovascular parameters (heart rhythm evaluation, respiratory sounds, peripheral oxygen saturation, and check for peripheral oedema) [174]. A detailed clinical history must include the full cardiovascular profile obtained in the acute COVID-19 phase, the infection severity as indicated by any newly arising short-term complications, and the treatments the patients has undergone since first symptom onset. Echocardiogram serves as a very useful tool in assessing any wall motion anomalies, indirectly indicating the status of the coronary vessel network.

New or persistent infectious myocardial involvement can be assessed through transthoracic echocardiography (TTE) or more advanced techniques such as cardiac magnetic resonance imaging (CMR) and positron emission tomography (PET) [175,176]. COVID-19 myocarditis treatment strategies involving steroid drugs, purified immunoglobulins, or antiviral therapy have yielded inconclusive results; newer protocols only recommend adapting previous strategies of treatment for viral myocarditis [177,178].

Life-threatening arrhythmias have been significantly correlated to age, basal heart rate, and antiviral therapy [179]. Atrial fibrillation is considered a significant occurrence in the acute phase of infections; new-onset arrhythmias are less encountered in the medium and long-term phases [180]. Drug–drug interactions explain the necessity of a thorough history regarding treatment directly targeted at COVID-19 infection [181]. Prolongation of QT corrected interval (QTc) through repeated electrocardiographic (ECG) recordings may help in identifying patients at risk for developing life-threatening arrhythmias [171,174,179].

Elevated concentration of N-terminal pro-B-type natriuretic peptide in a dyspnoeic post-acute COVID-19 patient may justify the requirement for TTE imaging [182,183]. A diagnosis of heart failure should be further managed according to the already existing guidelines [171]. Despite previous concerns regarding use of ACEI and possibly increased susceptibility of virus cell entry, current protocols encourage continuation of pre-existing treatment [184]. Echocardiography may view any abnormality in pulmonary hemodynamic, therefore predicting the course of acquired lung disease; integration of pulmonary status, right ventricle size and function, left ventricle function, and any valvular abnormality provide a more accurate image of cardiovascular and lung condition [185].

The increased risk for thrombotic events in the context of recent COVID-19 infection may be reversed by administering standard prophylactic anticoagulation: 40 mg enoxaparin daily [186]. Any higher dose did not yield a significant benefit regarding thrombotic versus haemorrhagic event occurrence in an inpatient setting. Outpatient pharmacological prophylaxis has been proposed, and guidelines provide consistent recommendations; severe cases of COVID-19 without any thromboembolic risk factor may benefit from extended anticoagulation, up to 6 months [174,187].

### 6.3. COVID-19-Related Chronic Kidney Disease Predictors

Acute kidney injury and chronic kidney disease have previously demonstrated a reciprocal correlation, one serving as a major risk factor for the development of the other [188]. Similarly, the relationship between COVID-19 severity and CKD course is, most probably, bidirectional [189]. At the cellular level, CKD is characterized by chronic inflammation, fibrosis, abnormal apoptosis rates, hypoxia, and vascular dysfunction, all processes which have been linked to medium-term consequences of COVID-19 [117,118,120,123,125]. The extent of decline in renal function directly determined by the amplitude of glomerular filtration rate drop during the AKI episode, the balance of regenerative and destructive processes, and, nonetheless, the pre-existing grade of CKD [190,191]. The risk for progression of CKD was evident even in COVID-19 patients who did not require hospitalization; expectedly, risk was highest among the admissions to critical care unit [192]. Oxygen therapy requirement in ICU is not, however, an entirely accurate predictor for rapidly worsening kidney function [193].

Inconsistency in reports regarding long-term kidney damage due to COVID-19 infection may be linked to an inconsistent protocol for CKD staging [194]. Overestimation of kidney damage was presumably attributed to an inadequate diagnostic procedure in the context of late-phase COVID-19 survivors [166,194]. Renal outcome in long-COVID patients should be compared with similar age-group kidney function, but also with CKD prevalence in general population. Proteinuria should be a key diagnostic criterion for COVID-19-related CKD, owing to the age-adapted definition of CKD, and the high frequency proteinuria is reported in COVID-19 patients [195]. These would allow for better identification of subjects at high-risk of rapid worsening renal function and could facilitate adequate medical care in the likelihood of increasing demand for intermittent dialysis [196].

## 7. Conclusions

SARS-CoV-2 is a currently running search for optimal management both for sick patients and for subjects who have contracted the disease. Renin-angiotensin-aldosterone system control an overwhelming range of processes at the cellular level. RAAS equilibrium disturbance is determined by the downregulation of ACE2 expression in the context of COVID-19 infection. The diversity and multitude of local effects controlled by this same axis comes to reinforce the strong connection between ACE2 alteration and organ-specific consequences of the disease. The brain, a central compartment exhibiting significant expression of the ACE2/Ang 1-7 axis, displays both local and central relay-controlled mechanisms. The cardiovascular and renal late manifestations of the disease are therefore also determined by a local and the central relay mechanism. There is an intricate network between these organs, and disruption of one will surely produce an echo in all others. Furthermore, there is rapidly emerging evidence concerning persistent or new-onset symptoms several months after COVID-19 infection was diagnosed. Knowledge regarding SARS-CoV-2 target organs and specific damage related to the infection, should enable healthcare providers to assess medical profile of long-COVID patients more precisely and promptly.

More refined protocols for diagnosis of long-COVID target-organ sequels should be elaborated. Brain damage secondary to pre-existing chronic diseases may sharply interfere with cerebral sequels related to long-COVID. Renal damage assessment should include at least one criterion to differentiate from function collapse related to aging—that is, proteinuria. Cardiovascular state evaluation should be thoroughly performed in high-risk categories of patients, in order to easily identify the mechanism for a potentially serious complication. The physician should obtain an anamnesis regarding in-hospital disease course, current and previously administered medication, should pay attention to subjective complaints of the patient, and shall promptly perform an electrocardiogram and echocardiogram.

The protective arm of RAAS influences a myriad of molecular level mechanisms: NO balance influencing, oxidative stress shielding, Na transmembrane transport, inflammation inhibition, apoptosis control and many more. These statements should encourage the scientists into developing more targeted treatments for COVID-19 survivors; we hope that new strategies of upregulating the levels of ACE2/Ang 1-7/MasR axis could prevent instalment of severe long-term consequences of the virus, or at least improve the quality of life for the already diagnosed patients several months post-infection. More research is required in order to clarify whether the ACE2 axis is the only fundamental cornerstone of long-term COVID-19 effects, yet it is nevertheless representative of the pathophysiology of the infection.

## Figures and Tables

**Figure 1 pharmaceutics-14-01906-f001:**
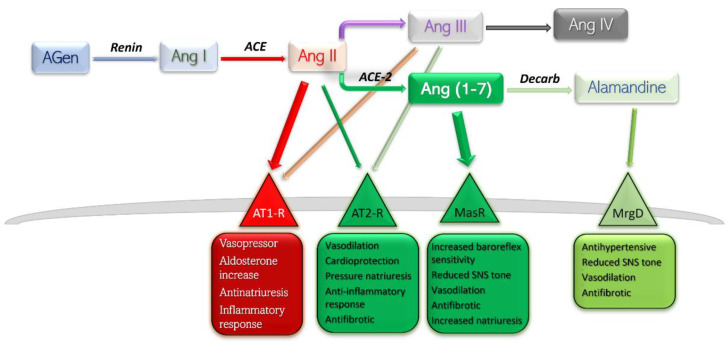
Schematic representation of RAAS components, their main receptors, and main actions. Deleterious side is shown in red-rimmed boxes; protective side is shown in green-rimmed boxes. Abbreviations: Agen, angiotensinogen; Ang, angiotensin; ACE, angiotensin-converting enzyme; Decarb, unspecified decarboxylases; MasR, Mas receptor pathway; MrgD, Mas-related G protein coupled receptor; AT, angiotensin receptor.

**Figure 2 pharmaceutics-14-01906-f002:**
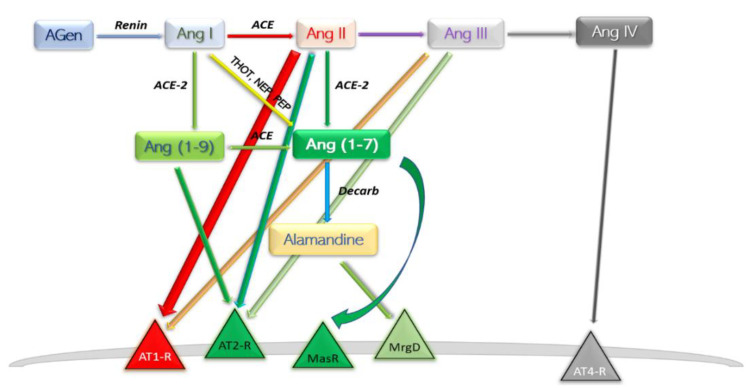
Schematic depiction of the modern view ACE2 axis and its products. AGen, angiotensinogen; Ang, angiotensin; ACE, angiotensin-converting enzyme; Decarb, unspecified aspartate decarboxylases; THOT, thimet oligopeptidase; NEP, neutral endopeptidase; PEP, prolyl endopeptidase; MasR, Mas axis; MrgD, Mas-related G protein receptor; AT–R, angiotensin receptor subtype.

**Figure 3 pharmaceutics-14-01906-f003:**
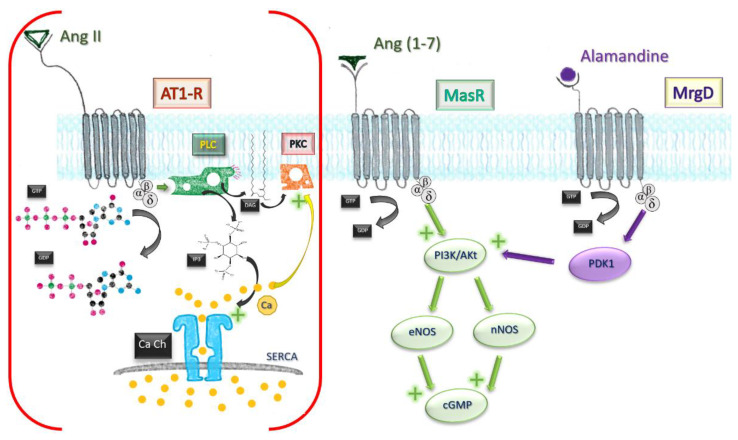
Comparative mechanisms of Ang II and Ang (1-7). Ang, angiotensin; MrgD, Mas-related G protein receptor D; AT1-R, angiotensin receptor type I; GTP, guanosine triphosphate; GDP, guanosine diphosphate; PLC, phospholipase C; PKC, protein kinase C; DAG, diacylglycerol; IP3, triphosphate inositol; Ca Ch, Calcium channel of smooth endoplasmic reticulum; eNOS, endothelial nitric oxide synthase; nNOS, neuronal; PI3K/Akt pathway; PDK1, phosphoinositide-dependent protein kinase 1.

**Figure 4 pharmaceutics-14-01906-f004:**
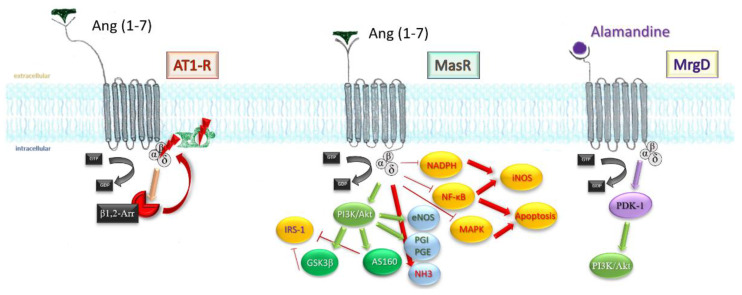
An extensive scheme displaying the mechanisms Ang (1-7) influences on an intracellular level. β-arr1/2, β-arrestins; GSK3β, glycogen synthase kinase 3β; AS160, Akt substrate 160; IRS-1, insulin receptor substrate 1; PGI, PGE, prostaglandins; NHE3, sodium hydrogen exchanger 3; NF-KB, nuclear transcription factor; iNOS, inducible nitrous oxide synthase; MAPK, mithogen-activated protein kinase; NADPH, nicotin-amide-diphospho-hydrogenase.

**Figure 5 pharmaceutics-14-01906-f005:**
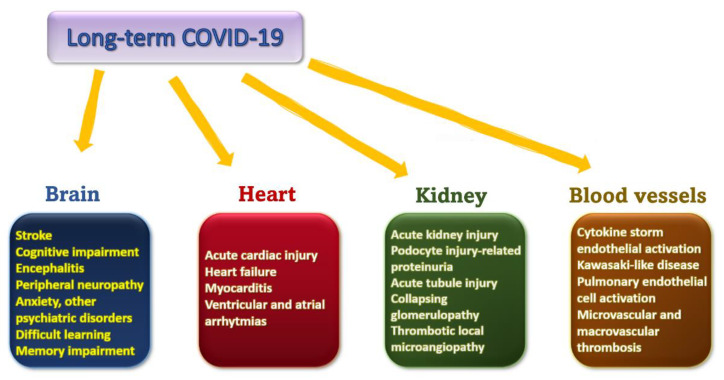
A summary of the previously described consequences of COVID-19 infection.

## Data Availability

Not applicable.

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
