# Peer review of "The Impact of Angiotensin-Converting Enzyme-2/Angiotensin 1-7 Axis in Establishing Severe COVID-19 Consequences"

_pharmaceutics, 2022, doi:10.3390/pharmaceutics14091906_

Round 1

Reviewer 1 Report

The review is well written and interesting to read. Some improvements are required to make it more reader friendly. Please refer to my comments below

A.     Figure 1: I suggest that the authors add on the figure the physiological consequences of the activation of each receptor (ie vasoconstriction, antinatriuresis, …)

B.     The title of fig 2 is confusinf, plase revise : Figure II. axis and its products. Also Figure IV. mediates its actions on an intracellular level.

C.     Please add a title for figure 3

D.    I suggest moving part 3.1.2. SARS-CoV-2 and its receptor, ACE2 just before paragraph 5. Medium and long-term complications determined by infection with SARS-CoV-2. This will create a link between the paragraphs about RAAS (before) and complications of COVID (After). It will be paragraph 4

E.     I suggest pooling paragraph 4 in the paragraph 3.3. since it refers to Ang1-7 so only one part of the paper speaks about ang1-7.

Example :

3.3. Angiotensin (1-7)

3.3.1. Intracellular signalling pathways employed by Angiotensin (1-7)

Add a sentence to introduce following parts such as: by touching Intracellular signalling pathways, Angiotensin (1-7) affect different physiological processes such as:

3.3.2. Metabolism

3.3.3. Oxidative stress

3.3.4. Arterial blood pressure

3.3.5. Apoptosis

These paragraphs will be subparagraphs of 3.3.1; Therefore 3.3.2 should be 3.3.1.1, etc… 3.3.5 should be 3.3.1.4

Change 4 Effects mediated by Angiotensin (1-7) in specific organs  to  3.3.2 Effects mediated by Angiotensin (1-7) in specific organs and subsequent paragraphs to 3.3.2.1 Brain etc…

F.     Change 6. COVID-19 infection severity. Proven risk factors and their link to ACE2 expression To 6. COVID-19 infection severity: Proven risk factors and their link to ACE2 expression

G.     Paragraphs 6.1.1 and 6.1.2 can be merged under the title 6.1 Age

H.    A lot of reviews highlighted complications of COVID infections, please cite :

1-     SARS-CoV-2, COVID-19, and Reproduction: Effects on Fertility, Pregnancy, and Neonatal Life

2-     The Pathophysiology of Long COVID throughout the Renin-Angiotensin System

Author Response

In reply to Reviewer 1,

We kindly appreciate your response, and we would like to thank for your time and patience assigned in reviewing this manuscript. Down below there is our response regarding your comments. We did consider most of your suggestions, and the revisions are marked in the revised version of the manuscript.

Starting with point A.

Owing to the reviewer’s suggestion, we decided on adding the most significant effects in a color-coded box for each of the substrate-receptor pairs. The new version of Figure I does not need any explanation down below, other than the used abbreviations. A short description of the classical view of RAAS was added to the text, just after Figure I. Also, in order to anticipate the Ang (1-7) – MasR and the Alamandine – MrgD influence on a larger-scale homeostasis, we added a short phrase. Therefore, the reader is presented in Figure I a very short resume of the effects Ang (1-7) and Alamandine mediate, actions which are going to be more deeply investigated further in the text. Please check revised version of manuscript.

Moving on to point B.

Title of Figure II, as well as a portion of the following block text were revised and completed accordingly. Please check revised version of manuscript.

Point C.

A purposeful title was added to Figure III. Please check revised version of manuscript.

Point D.

Owing to it`s specific cell entry mechanism, COVID-19 does alter the Renin-Angiotensin-Aldosterone System (RAAS) balance. The protective arm of the RAAS advances ACE2 as the most important member; downregulation of this enzyme would lead to general diminish of all components in this arm: lower concentrations of Angiotensin (1-7) and Ang (1-9). The link between the COVID patient and the lower Ang (1-7) and Ang (1-9) levels is clear.

We consider that introducing the COVID receptor mechanism in the 3.1.2 subparagraph would draw the readers attention specifically towards the end-products of reactions catalyzed by ACE2: namely, Ang (1-9) and Ang (1-7). It would represent the answer for the highly likely question “why these peptides specifically?” coming from readers. We did highlight the enzyme downregulation mechanism in order to specifically anticipate the reader`s interest in the effects mediated by those peptides, which are further explained in great detail in paragraphs 3.2 and 3.3.

Furthermore, every subparagraph of the section 5 (5.1. Brain, 5.2. Heart …) is met with constant reinforcement regarding the ACE2/ACE disequilibrium in COVID patient. To sum up, we consider that the mechanism of COVID cellular entry should remain at 3.1.2, owing for a better reading flow and understanding of concept.

Point E.

Changing the number for paragraph 4. to a subparagraph of the same 3.3. Ang (1-7) is a lot more sensible, as you have suggested. The revising is done accordingly. The following paragraph will also drop by 1: “5. Medium and long-term complications determined by infection with SARS-CoV-2” becomes now “4. Medium and long-term complications determined by infection with SARS-CoV-2”.

Point F.

The dot was replaced by a simple colon.

Point G.

Subparagraphs 5.1.1. and 5.1.2 were merged under the already existing "5.1. Age”.

Point H.

Please check revised version of manuscript. We added a new subchapter to the article, namely 5.6 Pregnancy. We consider that contents of this subchapter do fit in the aim of this review; the reader should benefit from acknowledging that pregnancy itself is a risk factor for contracting the infection, and that outcomes of pregnant patients do follow the mechanisms presented in previous sections of the review.

The review by Khazaal et al. was included in the beginning of newly added paragraph 6. Long-COVID organ-specific management. According to the suggestions received from Reviewer 2, we considered sensible to develop a new chapter, to describe the difficulties currently encountered by health care professionals in their attempt to diagnose long-COVID associated pathology. Besides, we addressed essential parameters for evaluating the presence of absence of specific complications, organ targeted. Finally, current management guidelines were mentioned.

Reviewer 2 Report

This article can be published with following revising

1. we reinforce the connection 25 between the local effects of RAAS and the noted consequences of COVID-19 "should be" replaced with identifying the gaps

2. Figure I can be replaced with better figure and more attractive that can attract audiences more

3. Figure II.  is not consistent with previosu figure

4. C an they add troponin information on a systematic way

5. Figure IV is consistent number, should follow either numerical or ...

6. figure IV background should be more visible

7. should discuss the consequence why, how and prevention 

8. Conclusion is not enough

9. Not sure why this paper discuss so many parameter without depth for each section

Author Response

In reply to Reviewer 2,

We kindly appreciate your response, and we would like to thank for your time and patience assigned in reviewing this manuscript. Down below there is our notes regarding your comments. We did consider each and every of your suggestions; the manuscript was revised and improved accordingly.

Point 1.

We consider that knowledge regarding both the physiological effect of ACE2 and the pathological condition of COVID-19 infection is readily available. The novelty and usefulness of this review is highlighting key pieces of information linking the two situations. The final part of the abstract was slightly revised according to your suggestion.

Point 2.

Figure I was improved with a resume of effects in a color-coded table under each substrate-receptor pair.

Point 3.

Figure II represents a more thorough view of the Renin-Angiotensin-Aldosterone System (RAAS). As mentioned in text, first figure depicts a more classical approach, while the second one is the modern view of the RAAS. Therefore, the two figures look seemingly different.

Point 4.

As per your advice, more information regarding different forms of Troponin – namely T and I – was added in section “4.2. Heart”. Correlation with in-hospital patient course was significant for both TnT and TnI, therefore it was worth mentioning in the main text. Three references were also added (113-115).

Point 5.

The figures now consistently follow roman numerals: I, II, III, IV and V.

Point 6.

The background in both Figure III and IV is a simplified view of a phospholipid double layer. We consider that the bilayer is not of interest to the reader, therefore a less visible color was chosen. Following your advice, we clearly stated to the left of the Figure IV which side represents the extra- and intra- cellular media.

Point 7.

Following your suggestions, an entirely new chapter was dedicated towards long-COVID consequences. Chapter 6. Long-COVID organ-targeted management was divided into three sections, for each organ - brain, cardiovascular system, and kidney. It is a sum of clinical based knowledge, commonly encountered long-term pathologies in COVID survivors, diagnostic difficulties, together with current therapy guidelines.

We decided to include this chapter just before conclusions, for two main reasons. First one, the patients most vulnerable to develop COVID-19 long-term consequences proved to be the same who present higher risk for severe COVID-19 infection; therefore, chapter 5 which describes the main high-risk categories of patients should be presented before 6. Secondly, a thorough description of the expected long-COVID symptoms and their associated mechanisms should occur prior to chapter 6; these are presented in chapters 3 and 4.

Please check revised manuscript.

Point 8.

The concluding paragraph was revised. Attention was slightly drawn towards earlier recognition of COVID-19 long-haul symptoms, and towards key practice aspects which would facilitate diagnosis. A brief enumeration of some of the processes controlled by Ang (1-7) was also added.

Point 9.

This review aims to highlight the striking link between ACE2 and long-term COVID-19 pathology, through a concise description of the actions mediated by Ang (1-7) at cellular level and the was COVID-19 indirectly influences this peptide. In elaborating this review, we aimed at developing a useful tool for healthcare providers, while focusing the readers` attention on the main player in COVID-19 pathology - ACE2. 

Reviewer 3 Report

The paper describes the effects of the CORONA virus infection on the ACE-2-related biological process. This review contains a concise and broad knowledge of the Ang1-7 effects on the body. It is an especially sound manuscript for readers, but one figure contains the wrong chemical structures. The following point should be checked and revised.

In Fig3, the chemical structure of GDP is wrong; one phosphate group should be diminished. The chemical structure of IP3 is also confusing. The chemical structure of IP3 in Figure 3 looks like trinitro-hexane; the phosphate position and structure of phosphate chemical groups should be correctly shown.

Author Response

In reply to Reviewer 3,

We kindly appreciate your response, and we would like to thank for your time and patience assigned in reviewing this manuscript. 

Figure III was thoroughly revised. A phosphate group was removed from GDP ball-stick structure. Inositol triphosphate structure was remade using professional software; oxygen at hydroxide substituents in phosphate groups are represented along with a negative charge, accounting for the common pH encountered inside the cell.

Round 2

Reviewer 1 Report

In general, some of the changes in the manuscript improved its quality; however the main weakness remain in the structure and the subparagraphs. There is no clear structure and logic in the text flow

The revised version contains still a lot of issues to be corrected.

Reviewer 2 Report

accept